# Effects of the *In Ovo* Administration of L-ascorbic Acid on the Performance and Incidence of Corneal Erosion in Ross 708 Broilers Subjected to Elevated Levels of Atmospheric Ammonia [note 1]

**DOI:** 10.3390/ani13030399

**Published:** 2023-01-24

**Authors:** Ayoub Mousstaaid, Seyed Abolghasem Fatemi, Katie Elaine Collins Elliott, April Waguespack Levy, William Wadd Miller, Hammad A. Olanrewaju, Joseph L. Purswell, Patrick D. Gerard, Edgar David Peebles

**Affiliations:** 1Department of Poultry Science, Mississippi State University, Starkville, MS 39759, USA; 2Poultry Research Unit, USDA-ARS, Starkville, MS 39762, USA; 3DSM Nutritional Products, Parsippany, NJ 07054, USA; 4Advanced Animal Eye Care, 3308 Old West Point Road, Starkville, MS 39759, USA; 5Department of Mathematical Sciences, Clemson University, Clemson, SC 29634, USA

**Keywords:** ammonia, broilers, eye lesion, performance, *in ovo* injection, L-ascorbic acid

## Abstract

**Simple Summary:**

Ammonia (NH_3_) is recognized as the most abundant toxic gas in poultry houses, and broilers subjected to elevated NH_3_ levels have exhibited negative effects on their growth, immunity, and respiratory systems. The objective of this research was to investigate the *in ovo* feeding of L-ascorbic acid (L-AA) on the post-hatch performance and corneal erosion incidence in Ross 708 broilers exposed to 50 ppm of atmospheric NH_3_. The 4 *in ovo* treatments that were applied at 17 days of incubation were: non-injected (control), saline-injected (control), or saline containing 12 or 25 mg of L-AA. It was found that the *in ovo* injection of 12 mg of L-AA improved live performance and lowered corneal erosion incidence relative to those injected with saline. In conclusion, the *in ovo* administration of 12 mg of L-AA may overcome the negative effects of high atmospheric NH_3_ concentrations on broiler live performance and corneal erosion incidence. Further research is needed to determine the physiological and immunological mechanisms that may be involved in the aforementioned improvements in broilers subjected to chronic exposures to elevated levels of atmospheric NH_3_.

**Abstract:**

Effects of the *in ovo* injection of various levels of L-ascorbic acid (L-AA) on the performance and corneal erosion incidence in Ross 708 broilers exposed to 50 parts per million (ppm) of atmospheric ammonia (NH_3_) after hatch were determined. A total of 1440 Ross 708 broiler embryos were randomly assigned to 4 treatments: non-injected (control), 0.85% sterile saline-injected (control), or saline containing 12 or 25 mg of L-AA. At hatch, 12 male chicks were randomly assigned to each of 48 battery cages with 12 replicate cages randomly assigned to each treatment group. All birds were exposed to 50 ppm of NH_3_ for 35 d and the concentration of NH_3_ in the battery cage house was recorded every 20 s. Mortality was determined daily, and mean body weight (BW), BW gain (BWG), average daily BW gain (ADG), and feed intake, as well as feed conversion ratio (FCR), were determined weekly. From 0 to 35 d of post-hatch age (doa), six birds from each cage were selected and sampled for eye erosion scoring. Incidences of corneal erosion were significantly higher at 21 and 28 doa in comparison to those at 14 and 35 doa, and at 21 doa, birds in the saline-injected group exhibited a higher incidence of corneal erosion compared to all other treatment groups. The *in ovo* injection of 12 mg of L-AA increased BWG (*p* = 0.043) and ADG (*p* = 0.041), and decreased FCR (*p* = 0.043) from 0 to 28 doa in comparison to saline-injected controls. In conclusion the *in ovo* administration of 12 mg of L-AA may have the potential to improve the live performance of broilers chronically exposed to high aerial NH_3_ concentrations, but further study is needed to determine the physiological and immunological factors that may contribute to this improvement.

## 1. Introduction

Antioxidant activity and immune function of broilers has been shown to promote in response to L-ascorbic acid (**L-AA**) [1]. Additionally, L-AA has been observed to improve immunoresponsiveness and increase disease resistance in poultry by enhancing their immune system [2,3,4]. Many animals and plants are able to synthesize L-AA [5]. It is well documented that many species, including poultry, are able to synthesis L-AA, while guinea pigs, non-human primates, and some birds are unable to synthesize L-AA in either the kidney or liver [6,7]. Thaxton and Siegel [8,9] further reported that L-AA has the ability to protect the immune tissues of growing birds and to subsequently reduce their mortality when they are infected with the infectious bursal disease virus. Additionally, L-AA synthesis in chickens has been documented to be poor under stressful conditions, such as low or high environmental temperatures, high ammonia (**NH_3_**) levels, high egg production rates, and parasitic infestations [10,11].

*In ovo* administration is considered an efficacious alternative approach for the posthatch vaccination of broilers. Previous research has shown that vaccine delivery into the amnion or embryo body proper are optimal sites for *in* ovo vaccination [12,13,14]. *In ovo* Injection is commercially used to administer particular vaccines between 17.50 and 19.25 d of incubation (**doi**) [14]. *In ovo* injection allows for uniform vaccine delivery into each injected egg, poses a limited contamination risk, and stimulates an early immune response in broilers [13,14,15,16,17,18,19]. *In ovo* administration is also less labor intensive and is relatively less stressful for the embryo when compared with the more stressful manual vaccination of hatchlings [13]. *In ovo* injection against Marek’s disease has been commercialized by the industry, whereas several researchers have conducted research to determine the effects of the *in ovo* administration of various nutrients such as L-ascorbic acid [20], vitamin D_3_ metabolites [16], and glucose [15]. 

Broiler housing requires adequate adaptive measures to avoid various implications associated with atmospheric contaminants such as NH_3_, carbon dioxide, and dust. At high levels of accumulation, these gases can reach toxic levels. In a commercial poultry house, NH_3_ concentrations can exceed 100 parts per million (**ppm**), particularly when there is no proper ventilation. Some of these gases that emanate from manure can build up rapidly to levels that are harmful to broilers [21]. For broilers, NH_3_ has been reported to act as a potential stressor, and at elevated levels has been shown to result in a decline in performance, an increase in inflammation, and a susceptibility to pathogenic agents [22,23]. The effects of NH_3_ exposure on broiler performance has been well investigated [24,25,26,27,28]. An exposure of 20 to 70 ppm of NH_3_ has been shown to reduce the growth and damage the respiratory system of broilers [29,30,31]. It is also well documented that high NH_3_ levels can cause corneal damage. Inflammation of the cornea and the secretion of corneal conjunctiva in chickens is called keratoconjunctivitis [32]. Greater than 50 ppm of NH_3_ has been reported in modern poultry houses, and 200 ppm of NH_3_ has been recorded in poorly ventilated facilities [29]. Moreover, Quarles et al. [33] observed that viable bacterial particle populations per cubic foot of air increased when NH_3_ was measured between 25 and 50 ppm. Levels of NH_3_ at 25 and 50 ppm also caused significant increases in airsacculitis in Leghorn chicks [34].

In humans, the corneal inflammation has also been shown to reduce in response to supplemental L-AA [35]. Brubaker et al. [36] isolated 1.33 mg of L-AA in each gram of wet corneal tissue in humans. Recognizing that L-AA is an anti-stress agent and that it helps improve the growth performance of broilers [20,37], it may also be beneficial to birds exposed to elevated aerial NH_3_ concentrations. Effects of the *in ovo* injection of L-AA on eye lesion incidence and the performance of broilers subjected to high atmospheric levels of NH_3_ have not been previously reported. It is hypothesized that some level of the *in ovo* injection of L-AA would lower the negative effects of chronic aerial ammonia exposure on the live performance of broilers. Therefore, the objectives of the current study were to determine the effects of the *in ovo* administration of L-AA on the growth performance and incidence of eye erosion in Ross 708 broilers exposed to elevated atmospheric levels of NH_3_ from 0 to 35 d of posthatch age (**doa**).

## 2. Materials and Methods

### 2.1. Egg Incubation

Ross 708 broiler hatching eggs from a 35-wk-old commercial broiler breeder flock were used and stored under commercial conditions (12.8 °C and 10.4 °C dry and wet bulb temperatures, respectively) for 24 h according to the procedure described by Fatemi et al. [38]. Eggs were then kept at room temperature for 4 h before being set. Sixty-five eggs were set in each of 4 treatment groups that were randomly arranged on each of 6 replicate tray levels (blocks) (1560 total eggs) in a Chick Master single-stage incubator (Chick Master Incubator Company, Medina, OH, USA) set in the setter at 37.5 °C dry bulb and 29.0 °C wet bulb temperatures and in the hatcher at 36.9 °C dry bulb and 29.0 °C wet bulb temperatures. Incubator air temperature and relative humidity were monitored following the procedure presented by Fatemi et al. [39]. Eggs were turned 24 times daily between 0 and 17 doi [40]. All eggs were candled at 12 and 17 doi to remove infertile eggs and those that did not contain live embryos. Furthermore, the difference between mean egg weight prior to set and mean egg weight at 12 and 17 doi was used in determining mean percentage egg weight loss (**PEWL**) for each treatment-replicate group between 0 and 12, 12 and 17, and 0 and 17 doi. Mean PEWL was calculated according to the procedure of Peebles et al. [41], and PEWL between 12 and 17 doi was determined after removal of eggs that did not contain live embryos at 12 doi. 

According to the procedure of by Zhang et al. [20], fresh 100 μL solution volumes of each pre-specified treatment were prepared at 17 doi prior to injection, in which 0.1 mL of saline (0.85% NaCl) contained 12 or 25 mg of AA. The 4 pre-specified treatments were: a non-injected control, the sham injection of a 100 μL volume of 0.85% sterile saline, or the injection of a 100 μL volume of sterile saline containing either 12 (**L-AA 12**) or 25 (**L-AA** 25) mg of L-AA. An Inovoject m multi-egg injection machine (Zoetis Animal Health, Research Triangle Park, NC, USA) was used to administer the *in ovo* injection that targeted the amnion. L-Ascorbic acid (A92902, Sigma-Aldrich Inc., St. Louis, MO, USA) solutions were prepared according to the method described by Zhang et al. [20]. Additionally, one egg from each of the 4 *in ovo* treatment groups on each of the 6 replicate tray levels (24 total eggs) were injected with Coomassie brilliant blue G-250 (colloidal) dye for embryo staging analysis according to the procedure described by Sokale et al. [42].

### 2.2. Hatch Residue, Hatchability, and Posthatch Performance

Hatch residue analysis was conducted as described by Ernst et al. [43] to determine post-injection embryonic mortality. Late, pip, post-pip, and hatchling mortalities were described, respectively, as those mortalities that occurred between 17 and 21 doi prior to pip, during the pipping process, after the pipping process, and immediately after complete emergence from the shell. Furthermore, hatchability of injected live embryonated eggs (**HI**) was determined at 21 doi. Residue and HI values were calculated as percentages of injected eggs that contained live embryos of 17 doi. After hatch, mean hatchling body weight (**BW**) of the chicks belonging to each replicate group in each treatment was determined by dividing total chick BW by the numbers of chicks in each group. All chicks were feather-sexed to select for male broilers in their prespecified treatment, and then male chicks from each replicate basket belonging to a common treatment were pooled. Twelve randomly selected male broilers from the pool were placed in each of 12 replicate battery cages belonging to each treatment group (12 replicates × 4 *in ovo* injection treatments = 48 total cages). The cages assigned to the various replicate-treatment groups were randomly arranged throughout a common battery room, with an empty cage between those that were occupied to avoid the contact of chicks from different treatments. 

All birds received a Mississippi State University basal corn-soybean diet (Table 1) formulated to meet Ross 708 commercial guidelines [44,45]. Birds were provided feed and water for ad libitum consumption throughout the 35 doa, and from 0 to 14 doa and from 15 to 35 doa, starter and grower diets were, respectively, fed. Mean live performance variables of the birds including BW, BW gain (**BWG**), average daily gain (**ADG**), feed intake (**FI**), and average daily feed intake (**ADFI**) were obtained in the 0 to 7, 8 to 14, 15 to 21, 22 to 28, 29 to 35, 0 to 14, 15 to 28, 0 to 28, and 0 to 35 posthatch doa phases of growout. Percentage mortality and feed conversion ratio (**FCR**; g of feed/g of gain), which was adjusted for bird mortality, were calculated for the same time periods. At 28 doa, one bird from each of 12 replicate cages in each treatment group (48 total birds) was individually weighed and the absolute weights of their pectoralis major (**P. major**) and minor (**P. minor**) muscles were determined. The relative weights of the P. major and P. minor muscles, as percentages of BW, were reported. Additionally, absolute breast muscle weight was calculated as the sum of the P. major and P. minor weights, and its relative weight was also reported as a percentage of BW.

### 2.3. Ammonia Exposure 

To eliminate the possibility of the release of additional atmospheric NH_3_ from used litter, birds were placed in suspended battery cages. Ammonia was released from a compressed NH_3_ gas tank (NexAir, LLC, Memphis, TN, USA) and gas flow was controlled with a solenoid valve triggered by a process controller (Precision Digital Corp., Hopkinton, MA, USA). Pipes (PVC) were used to evenly distribute and sample the NH_3_ gas throughout the battery room. A 1/20 HP Diaphragm Compressor/Vacuum Pump (Model # 107CAB18; W.W. Grainger Co., Lake Forest, IL, USA) was used to receive input from 6 different locations (4 corner and 2 medial) within the room and delivered these to a photoacoustic gas analyzer (Chillgard RT, Mine Safety Appliance, Cranberry Township, PA, USA) that monitored the composite NH_3_ sample. Ammonia gas diffusion and metering was accomplished with the use of a metering device connected to the gas tank. Moderate and constant ventilation was available in the room to further simulate an industry-like environment. Positive pressure ventilation was provided by an EX-Belt Drive Fan (Model: DDP12B4 115/230V; Dyna Masters Inc., Rochester, NY, USA) and a Blower Motor (Mod No:5K907D; Dayton Manufacturing Co., Dayton, OH, USA) positioned in both sides of the battery cage room. Fans were used to limit high NH_3_ concentration, as the solenoid valve released NH_3_ when levels reached 45 ppm, and the fans brought in fresh air when NH_3_ levels reached 65 ppm. Throughout the 5-wk period that the broilers were exposed to the atmospheric NH_3_, the average NH_3_ concentration in the battery room approximated the designated level of 50 ppm. The recorded NH_3_ levels ranged between 42.5 and 49.7 ppm (Table 2).

### 2.4. Eye Corneal Evaluation

At the beginning of the trial, 2 birds from each of the 12 replicate cages per treatment were randomly selected and permanently tagged for their identification and repeated corneal erosion evaluation at 0, 7, 14, 21, 28, and 35 doa. Ocular examinations were conducted by a single board-certified veterinary ophthalmologist who was blinded as to treatment category. Biomicroscopy was performed using a Kowa SL-14 portable slit lamp (Kowa Co., Sakai, Osaka, Japan). Corneal erosion grading was similar to the classifications previously introduced by Miles et al. [46]. The numerical scale for grading of corneal erosion was: 0 = normal; 1 = shows signs of ocular inflammation; and 2 = corneal perforation/lesions and keratoconjunctivitis.

### 2.5. Statistical Analysis

The experimental design was a randomized complete block for both the incubational, hatch, and posthatch rearing periods. Incubator level in the setter and hatcher served as the unit of treatment replication for the hatch data, and battery cage as the unit of treatment replication for the performance, meat yield, and eye erosion scoring data. All incubational, hatch, and posthatch performance data within each individual time period were separately analyzed by ANOVA. All data was analyzed using the procedure for linear mixed models (PROC GLIMMIX) of SAS 9.4^©^ [47] that employed the following model [45,48]
Yij = μ + Bi+ Tj + Eij(1)
where μ was the population mean; Bi was the block factor (i = 1 to 12); Tj was the effect of each *in ovo* injection treatments (j = 1 to 4); and Eij was the residual error. 

A two-way repeated measures ANOVA was used to analyze the eye erosion score data. Fisher’s protected least significant difference [49] was used to separate means, and differences between means were designated significant at *p* ≤ 0.05. 

## 3. Results

### 3.1. Hatch and Broiler Performance

The sites of injection across treatment were confirmed to be 93.4% and 6.6% in the amnion and embryo body proper, respectively. No significant *in ovo* treatment differences were observed for 0 to 12, 12 to 17, and 0 to 17 doi PEWL, or for hatchling BW and HI (Table 3). Hatch residue analysis also revealed that late, pip, post-pip, and hatchling mortalities did not significantly differ among *in ovo* treatments (Table 3). Birds in the non-injected and L-AA 12 *in ovo*-injected treatment groups had a significantly higher BW at 28 doa in comparison to those in the saline-injected group (Table 4). Conversely, no significant *in ovo* treatment differences were observed for the other posthatch performance variables of the birds, including BWG, ADG, FI, and ADFI from 0 to 7, 8 to 14, 15 to 21, 22 to 28, 29 to 35, 0 to 14, and 0 to 35 doa. There was also no significant treatment effect on FCR between 8 and 14, 15 and 21, 29 and 35, 0 and 14, 15 and 28, and 0 and 35 doa; on FI and ADFI between 15 and 28, and 0 and 28 doa; or on ADG between 15 and 28 doa. However, birds in the L-AA 12 treatment group had a lower or improved FCR between 0 and 7, 22 and 28, and 0 and 28 doa in comparison to saline-injected controls. Furthermore, BWG between 15 and 28 and 0 and 28 doa, and ADG between 0 and 28 doa were greater in birds belonging to the L-AA 12 treatment in comparison to those in the saline-injected control group (Table 4).

For the individually selected birds sampled for eye collection, there was no significant treatment difference for their BW at 7, 14, 21, and 35 doa. In addition, at 0, 7, 14, 21, and 28 doa, there were no significant treatment differences in absolute eye weight and eye weight as a percentage of BW (Table 5). Likewise, no significant treatment differences were observed for the absolute or relative (percentage of BW) weights of the P. major, P. minor, and breast muscle at 28 doa (Table 6).

### 3.2. Eye Lesion Scoring

There were no eye lesions observed prior to the second wk of the posthatch period, and no category 2 scores were observed and recorded for any of the birds evaluated. Therefore, only the category 0 and 1 score data at 14, 21, 28, and 35 doa are presented in Table 7. For category 0 and 1 corneal erosion scores, there was a significant interaction between *in ovo* injection treatment and doa. At 21 doa, the saline-injected group had a greater percentage of 1 and lower percentage of 0 corneal erosion scores compared with all the other treatment groups. However, at 14, 28, and 35 doa, no significant treatment differences were observed for percentages of either 0 or 1 scores (Table 7).

## 4. Discussion

The objectives in this study included determinations as to the effects of the *in ovo* administration of different levels of L-AA on the hatch results of broilers. The lack of a treatment effect on the PEWL results in all the doi intervals examined showed that all eggs across treatment-replicate group were exposed to similar environmental conditions, including temperature and humidity, through 17 doi. Previous studies [38,39,50] have likewise used PEWL as a means to confirm equal incubational conditions among treatment groups in an incubator, thereby eliminating environment as a confounding factor. Mousstaaid et al. [51] previously found that the *in ovo* feeding of 12 mg of L-AA decreased late embryonic mortality. However, no significant treatment differences were observed in the hatch residue analysis of the current study (Table 3). These current results are similar to those reported by Zhang et al. [20] and Mousstaaid et al. [52]. Zhang et al. [20] showed no effect of the amniotic injection of 0.5 to 13.5 mg of L-AA on the subsequent hatch residue analysis results. The subsequent Mousstaaid et al. [52] likewise noted no effect on the amniotic injection of 12 or 25 mg of AA on their hatch residue results. There are several possible reasons for contrasts in the hatch results of these studies. Procedural differences in the studies that may have led to these contrasts include differences in incubator type, the timing of *in ovo* injection, and the level of L-AA injected.

The *in ovo* feeding of different levels of L-AA has shown promising results concerning the overall live performance of broilers under normal rearing conditions. The *in ovo* feeding of 3 to 6 mg of L-AA at 17 doi enhanced the posthatch performance (BW, ADG, and ADFI) of Ross 708 broilers [37]. Additionally, the *in ovo* administration of L-AA at 3 mg in Arbor Acres broiler hatching eggs at 11 doi has also been shown to result in an increase in their ADG and a decrease in their FCR from 0 to 21 doa [52]. Furthermore, in a previous companion study in our laboratory, it was found that the *in ovo* injection of 25 mg of L-AA decreased FCR in the first week posthatch in comparison to a saline-injected control group. It is well documented that the *in ovo* administration of lower levels of L-AA have positive effects on posthatch broiler performance after 14 doa [37,53,54], and that the *in ovo* administration of L-AA has improved posthatch broiler performance by increasing ADG and ADFI from 0 to 41 doa [52]. 

Effects of L-AA administered dietarily or by *in ovo* injection on the live performance of chickens during elevated atmospheric NH_3_ exposure have not been previously investigated. It was hypothesized that the *in ovo* injection of L-AA would reduce the incidence of detrimental ocular changes caused by high atmospheric NH_3_ levels. Therefore, further objectives in current research were to investigate the effect of these *in ovo* treatments on the posthatch performance as well as the incidences of ocular erosion in broilers exposed to 50 ppm of atmospheric NH_3_ from 0 to 35 doa. In comparison to the saline-injected treatment group during the NH_3_ challenge, the *in* ovo injection of L-AA 12 improved FCR from 0 and 7, 22 to 28, and 0 to 28 doa, and increased BWG from 15 to 28 and 0 to 28 doa. Because L-AA may mitigate the impacts of changes in immune function, inflammatory response, and tissue oxidation levels in broilers subjected to elevated atmospheric NH_3_ levels, these 3 physiological properties are addressed as potential means by which the vision and subsequent overall performance of the birds in this study were affected.

Effects of dietary or *in ovo* supplementation of L-AA on any of the immune variables of chickens subjected to NH_3_ gas have not been previously reported. However, the use of supplemental dietary L-AA has shown promising results for improving systemic humoral [55,56] and cell-mediated immune [1] responses in chickens. Kennes et al. [57] also reported that supplemental L-AA provided to elderly human patients improved their cell-mediated immune responses. It is well documented that the exposure of chickens to 30 to 70 ppm of atmospheric NH_3_ resulted in a depressed humoral immune response [27,58,59]. When broilers were subjected to 52 ppm of aerial NH_3_ for 3 wk, it has been shown in an earlier report that decreased levels of circulating IgA, IgG, and IgM subsequently occurred [57]. Complement component 4 (C4) plays an important role in tissue regeneration. Furthermore, lower levels of C4 are associated with immunosuppression [27]. When subjected to 30 ppm of NH_3_ for 25 wk, layers have exhibited lower plasma IgM as well as C4 concentrations [27]. Dietary supplementation of 200 ppm of L-AA has been observed to increase the number of CD4 (cluster of differentiation 4) cells and T-cell receptors (TCR-II), which reflects an increase in lymphocytes elicited in response to exogenous antigens [60]. Additionally, the *in ovo* injection of 3 mg of L-AA has enhanced the immune system by increasing levels of glutathione peroxidase and superoxide dismutase [61] and has lowered pro-inflammatory cytokines [1] in the spleen of broilers. Although the role of L-AA in lymphocyte activity is not clear, studies have shown it to be involved in the immunomodulatory function of lymphocytes, which affects the expression of immunoglobulins [62]. The beneficial effects of *in ovo* L-AA 12 on posthatch performance may therefore be in part due to an improvement in the local immune response of the broilers.

It is well documented that the exposure of broilers to atmospheric NH_3_ ranging between 30 and 70 ppm resulted in increases in their systemic inflammatory reactions [27,58,59]. Lower levels of C4 are also associated with higher levels of inflammation [63]. Layers that were subjected to 30 ppm of NH_3_ for 25 wk have exhibited lower plasma C4 concentrations, indicating that they had a lower level of protection against systemic inflammatory reactions [27]. Nitric oxide is considered as a pro-inflammatory agent when produced in excess under abnormal physiological conditions. Previously, it was observed in our laboratory that serum nitric oxide concentrations were lower in birds that had received *in ovo* injections of 12 mg of L-AA in comparison to those injected with saline. Other studies have also showed that supplemental dietary L-AA reduced inflammatory reactions [1,61,64] in broilers. The expression of IL-1β has been shown to increase in broilers subjected to 70 ppm of atmospheric ammonia [59]. However, the *in ovo* injection of 3 mg of L-AA has been observed to decrease the expression of pro-inflammatory cytokines including IL-1β, IL-6, and TNF-α [61]. The *in ovo* injection of 3 mg of L-AA has also increased initial antibody titers against Avian Influenza in both male and female ducklings and secondary antibody titers against Avian Influenza in female ducklings [64]. The noted improvements in broiler performance from 0 to 28 doa in response to 12 mg of *in ovo* L-AA in this study could be in part due to an improvement in their inflammatory response.

Another reason for the improvement could be due to an increase in the systemic or local antioxidant capacity of the broilers in response to the *in ovo* injection of L-AA. Commercial broilers are subjected to many stressors including high ambient temperatures [65,66], stocking densities [67], and high atmospheric NH_3_ challenges [68]. It is well documented that *in ovo* or dietary administration of L-AA can enhance the antioxidant activity of broilers [37,69]. Additionally, our laboratory previously found that in association with an increase in BWG, the *in ovo* injection of L-AA at 12 or 36 mg increased plasma concentrations of total superoxide dismutase and deceased the plasma concentrations of malondialdehyde, considering enzymatic and non-enzymatic antioxidant capacity factors, respectively, in broilers [37]. Furthermore, an increase in antioxidant activity is associated with a lowering of systematic and local oxidative stress [70,71]. 

Oxidative stress in the cornea can also stimulate the production of pro-inflammatory cytokines, proteolytic enzymes, and enzymes that synthesize nitric oxide [72]. Increasing and preserving antioxidant concentrations in the eye will help protect and maintain the homeostasis of ocular tissue [73]. Moreover, in a study conducted by Zhang et al. [37], which is a companion to this one, the *in ovo* injection of 12 mg of L-AA resulted in a reduction in the tracheal histomorphological lesions of broilers exposed to 50 ppm of atmospheric NH_3_ at 21 doa. In comparison to saline-injected controls, corneal erosion incidence was reduced at 21 doa when the birds had received an L-AA 12 or L-AA 25 treatment at 17 doi. Therefore, in the current study, the reduction in ocular lesion incidence at 21 doa in response to the *in ovo* injection of 12 or 25 mg of L-AA at 17 doi could be partially due to an improvement in tissue antioxidant activity. Furthermore, the promising results observed in live performance could also be partly due to a systemic improvement in the antioxidant activity of the broilers provided *in ovo* injections of L-AA. Because the current study is a sequel of the study conducted by Zhang et al. [37], in which the same *in ovo* injection and brooding management procedures were followed, it was expected that similar antioxidant activity effects would also occur in this study.

The eye evaluation results of the current study revealed that when broilers were exposed to 50 ppm of aerial NH_3_ from 0 to 35 doa, higher corneal erosion incidences were observed at 21 and 28 doa compared to those at 14 and 35 doa. Likewise, the exposure of broilers to 50 and 75 ppm of NH_3_ through 49 doa has been shown to result in higher incidences of corneal erosions from 14 to 28 doa [46], and birds subjected to atmospheric NH_3_ at 50 ppm from 0 to 14 doa has also been shown to cause higher incidences of corneal erosions at 14 doa [30]. Improvements in the performance of the broilers in response to the *in ovo* injection of L-AA while being exposed to high atmospheric NH_3_ concentrations may be related to the long posthatch period examined. These data indicate that at least 2 wk are needed for ocular erosions to become manifest when birds are exposed to aerial NH_3_ levels ranging from 50 to 75 ppm. On the other hand, corneal erosion incidence was reduced or eliminated after a continual 4 wk exposure to atmospheric NH_3_. This indicates a physiological capability of the birds to reverse the effects of NH_3_ on corneal damage with or without being previously provided *in ovo* injections of L-AA. Posthatch performance responded more favorably to the *in ovo* injection of 12 mg of L-AA rather than to 25 mg of L-AA. Similar results were observed when L-AA above 12 mg was administrated by *in ovo* injection. It was observed that the *in ovo* injection of 36 mg of L-AA negatively affected the BW and FCR of broilers in comparison to 12 mg L-AA and saline control groups [37]. Zhang et al. [37] suggested that the higher level of L-AA could lead to an inferior feed efficiency or to a subsequently poorer utilization of feed. Thus, this could be a partial reason for the unexpected response to the *in ovo* injection of 25 mg of L-AA. 

## 5. Conclusions

In conclusion, effects of the *in ovo* injection of L-AA on the performance and corneal erosion incidence of broilers exposed to 50 ppm of atmospheric NH_3_ was investigated. The results of the current study revealed that the *in ovo* injection of 12 mg of L-AA resulted in an improvement in broiler performance from 0 to 28 doa as well as resulting in a reduction in the incidences of corneal erosions at 21 doa. These improvements in response to the *in ovo* administration of L-AA may be linked to an increase in the local immune response and antioxidant activity, and a reduction in the inflammatory response of the birds, which has also been observed in related studies. However, more research is needed to determine the physiological and morphological mechanisms involved in improvements in the performance and corneal conditions of broilers in response to various levels of *in ovo* L-AA while being subject to elevated atmospheric NH_3_ levels during the rearing period.

## Figures and Tables

**Table 1 animals-13-00399-t001:** Feed composition of the experimental diets from 0 to 35 d of age (doa).

		Commercial Diet
		Starter (0 to 14 doa)
Item		
Ingredient (%)		Pct
	Yellow corn	53.23
	Soybean meal	38.23
	Animal fat	2.6
	Dicalcium phosphate	2.23
	Limestone	1.27
	Salt	0.34
	Choline chloride 60%	1
	Lysine	0.28
	DL-Methionine	0.37
	L-threonine	0.15
	Premix ^1^	0.251
	Coccidiostat ^2^	0.05
	Total	100
Calculated nutrients		
	Crude protein	23
	Calcium	0.96
	Available phosphorus	0.48
	Apparent metabolizable energy (AME; kcal/kg)	3000
	Digestible Methionine	0.51
	Digestible Lysine	1.28
	Digestible Threonine	0.86
	Digestible total sulfur amino acids (TSAA)	0.95
	Sodium	0.16
	Choline	0.16
		Grower (15 to 35 doa)
Item		
Ingredient (%)		Pct
	Yellow corn	57.13
	Soybean meal	34.8
	Animal fat	3.5
	Dicalcium phosphate	2
	Limestone	1.17
	Salt	0.34
	Choline chloride 60%	0.1
	Lysine	0.21
	DL-Methionine	0.32
	L-threonine	0.16
	Premix	0.25
	Coccidiostat	0.05
	Total	100
Calculated nutrients		
	Crude protein	21.5
	Calcium	0.87
	Available phosphorus	0.435
	AME (kcal/kg)	3100
	Digestible Methionine	0.47
	Digestible Lysine	1.15
	Digestible Threonine	0.77
	Digestible TSAA	0.87
	Sodium	0.16
	Choline	0.16

^1^ The broiler premix provided per kilogram of diet: vitamin A (retinyl acetate), 10,000 IU; cholecalciferol, 4000 IU; vitamin E (DL-α-tocopheryl acetate), 50 IU; vitamin K, 4.0 mg; thiamine mononitrate (B_1_), 4.0 mg; riboflavin (B_2_), 10 mg; pyridoxine HCL (B_6_), 5.0 mg; vitamin B_12_ (cobalamin), 0.02 mg; D-pantothenic acid, 15 mg; folic acid, 0.2 mg; niacin, 65 mg; biotin, 1.65 mg; iodine (ethylene diamine dihydroiodide), 1.65 mg; Mn (MnSO_4_H_2_O), 120 mg; Cu, 20 mg; Zn, 100 mg, Se, 0.3 mg; Fe (FeSO_4_.7H_2_O), 800 mg. ^2^ Decocx ® (Zoetis, Parsippany, Morris County, NJ, USA).

**Table 2 animals-13-00399-t002:** Average levels of ammonia (**NH_3_**) recorded in the battery cage room in the 0 to 7, 8 to 14, 15 to 21, 22 to 28, and 29 to 35 d of posthatch age (**doa**) periods.

	0 to 7 doa	8 to 14 doa	15 to 21 doa	22 to 28 doa	29 to 35 doa
NH_3_ level (ppm)	42.5	46.9	44.8	47.2	46.7
Standard deviation	10.5	10.4	12.6	9.3	8.6

**Table 3 animals-13-00399-t003:** Effects of treatment [non-injected; saline-injected (**saline**); saline containing 12 mg of L-ascorbic acid (**L-AA 12**), or 25 mg of L-ascorbic acid (**L-AA 25**) on percentage egg weight loss (**PEWL**) between 0 and 12, 12 and 17, and 0 and 17 d of incubation (**doi**), hatchability of injected live embryonated eggs (**HI**), hatchling BW, and hatch residue analysis variables (late, pip, post-pip, and hatchling mortalities) at 21 d of incubation (**doi**).

Treatment	0 to 12 PEWL	12 to 17 PEWL	0 to 17 PEWL	HI	Late ^1^	Pip ^2^	Post-Pip ^3^	Hatchling ^4^	Hatchling BW (g)
	----------------------------------------------%-------------------------------------------------								
Non-injected ^5^	4.2	6.0	10.2	94.4	2.1	0.0	1.1	0.0	44.5
Saline ^6^	4.3	6.1	10.4	93.1	4.7	1.7	1.1	0.3	43.5
L-AA 12 ^7^	4.3	6.1	10.4	94.8	4.4	0.0	1.1	0.3	43.7
L-AA 25 ^8^	4.2	6.0	10.2	94.1	3.9	0.9	1.1	0.7	43.4
SEM ^9^	0.07	0.15	0.20	2.16	1.35	0.82	1.43	0.30	1.17
*p*-value	0.453	0.545	0.576	0.888	0.243	0.159	0.136	0.504	0.730

^1^ Mortality between 17 and 21 doi, prior to pip.; ^2^ Mortality during the pipping process.; ^3^ Mortality after the pipping process.; ^4^ Mortality immediately after complete emergence of hatchlings from the shell.; ^5^ Eggs that were not injected.; ^6^ Eggs that were injected with 100 μL saline at 17 doi.; ^7^ Eggs that were injected with 100 μL saline containing L-AA 12 at 17 doi.; ^8^ Eggs that were injected with 100 μL saline containing L-AA 25 at 17 doi.; N = Approximately 65 eggs in each of 12 tray replicate groups in each treatment were used for means calculations.; ^9^ Standard error of the mean.

**Table 4 animals-13-00399-t004:** Effects of treatment [non-injected; saline-injected (**saline**); saline containing 12 mg of L-ascorbic acid (**L-AA 12**), or 25 mg of L-ascorbic acid (**L-AA 25**) on Ross 708 broiler live performance throughout the rearing period.

	7 doa	------------------------ 0 to 7 doa ^7^--------------------------				
Treatment	BW (g)	BWG ^1^ (g)	ADG ^1^ (g)	FI ^1^ (g)	ADFI ^1^ (g)	FCR ^1^ (g/g)
Non-injected ^2^	112	65.3	9.3	101	14.5	1.56 ^a^
Saline ^3^	110	63.4	9.1	99	14.3	1.59 ^a^
L-AA 12 ^4^	116	68.8	9.8	100	14.3	1.45 ^b^
L-AA 25 ^5^	112	65.3	9.3	99	14.1	1.52 ^ab^
SEM ^6^	2.2	2.18	0.31	2.3	0.33	0.028
*p*-value	0.152	0.134	0.134	0.806	0.813	0.029
	14 doa BW	-----------------------8 to 14 doa ^8^------------------------				
		BWG (g)	ADG (g)	FI (g)	ADFI (g)	FCR (g/g)
Non-injected	329	216	30.9	300	42.8	1.39
Saline	324	213	30.5	297	42.4	1.39
L-AA 12	341	226	32.2	295	42.1	1.32
L-AA 25	331	218	31.2	302	43.1	1.38
SEM	10.3	8.6	1.23	9.6	1.37	0.054
*p*-value	0.441	0.541	0.547	0.889	0.891	0.475
	21 doa BW	----------------------15 to 21 doa ^9^------------------------				
		BWG (g)	ADG (g)	FI (g)	ADFI (g)	FCR (g/g)
Non-injected	662	334	55.6	507	84.6	1.53
Saline	644	320	53.4	507	84.5	1.59
L-AA 12	675	334	55.7	508	84.7	1.52
L-AA 25	670	340	56.7	513	85.6	1.53
SEM	17.0	13.0	2.16	11.3	1.89	0.055
*p*-value	0.308	0.475	0.480	0.933	0.932	0.569
	28 doa BW	-----------------------22 to 28 doa ^10^------------------------				
		BWG (g)	ADG (g)	FI (g)	ADFI (g)	FCR (g/g)
Non-injected	1225 ^a^	563	80	807	115	1.43 ^b^
Saline	1152 ^b^	509	73	818	117	1.64 ^a^
L-AA 12	1226 ^a^	551	79	794	113	1.45 ^b^
L-AA 25	1193 ^ab^	522	75	816	117	1.58 ^ab^
SEM	27.7	22.2	3.16	24.1	3.4	0.057
*p*-value	0.048	0.080	0.081	0.737	0.737	0.050
	35 doa BW	------------------------29 to 35 doa ^10^-----------------------				
		BWG (g)	ADG (g)	FI (g)	ADFI (g)	FCR (g/g)
Non-injected	1956	731	104	1071	153	1.56
Saline	1870	717	102	1109	158	1.78
L-AA 12	1867	642	92	1121	160	1.89
L-AA 25	1839	646	92	1121	160	1.77
SEM	73.6	84.7	12.10	54.1	7.7	0.227
*p*-value	0.436	0.615	0.616	0.767	0.768	0.531
	0 to 14 doa ^8^					
	BWG (g)	ADG (g)	FI (g)	ADFI (g)	FCR (g/g)	
Non-injected	281	20.1	401	28.6	1.42	
Saline	277	19.8	397	28.3	1.44	
L-AA 12	294	21.0	394	28.2	1.35	
L-AA 25	284	20.3	401	28.6	1.41	
SEM	10.1	0.66	11.0	0.75	0.046	
*p*-value	0.378	0.365	0.912	0.914	0.275	
	15 to 28 doa ^10^					
	BWG (g)	ADG (g)	FI (g)	ADFI (g)	FCR (g/g)	
Non-injected	896 ^a^	69	1314	110	1.59	
Saline	828 ^b^	64	1326	111	1.75	
L-AA 12	885 ^a^	68	1302	109	1.60	
L-AA 25	862 ^ab^	66	1330	111	1.68	
SEM	17.4	1.9	32.1	2.68	0.046	
*p*-value	0.050	0.058	0.830	0.830	0.087	
	0 to 28 doa ^10^					
	BWG (g)	ADG (g)	FI (g)	ADFI (g)	FCR (g/g)	
Non-injected	1178 ^a^	42.1 ^a^	1715	66	1.57 ^b^	
Saline	1105 ^b^	39.5 ^b^	1722	66	1.69 ^a^	
L-AA 12	1179 ^a^	42.1 ^a^	1697	65	1.56 ^b^	
L-AA 25	1146 ^ab^	40.9 ^ab^	1730	67	1.63 ^ab^	
SEM	27.5	0.98	38.7	1.49	0.053	
*p*-value	0.045	0.041	0.845	0.851	0.050	
	0 to 35 doa ^10^					
	BWG (g)	ADG (g)	FI (g)	ADFI (g)	FCR (g/g)	
Non-injected	1839	53	2786	90	1.72	
Saline	1822	52	2831	91	1.78	
L-AA 12	1821	52	2818	91	1.76	
L-AA 25	1792	51	2851	92	1.80	
SEM	72.5	2.1	69.0	2.2	0.074	
*p*-value	0.934	0.914	0.817	0.887	0.728	

^a,b^ Treatment means within the same variable column within type of treatment with no common superscript differ significantly (*p* < 0.05).; ^1^ BW gain (BWG), average daily gain (ADG), feed intake (FI), average daily feed intake (ADFI), and feed conversion ratio (FCR); ^2^ Eggs that were not injected.; ^3^ Eggs that were injected with 100 μL saline at d 17 of incubation (doi).; ^4^ Eggs that were injected with 100 μL saline containing L-AA 12 at 17 doi.; ^5^ Eggs that were injected with 100 μL saline containing L-AA 25 at 17 doi.; ^6^ Standard error of the mean.; ^7^ N = 12 birds in each of 12 replicate groups in each treatment combination were used for means calculations.; ^8^ N = 10 birds in each of 12 replicate groups in each treatment combination were used for means calculations.; ^9^ N= 8 birds in each of 12 replicate groups in each treatment combination were used for means calculations.; ^10^ N = 6 birds in each of 12 replicate groups in each treatment combination were used for means calculations.

**Table 5 animals-13-00399-t005:** Effects of treatment [non-injected; saline-injected (**saline**); saline containing 12 mg of L-ascorbic acid (**L-AA 12**), or 25 mg of L-ascorbic acid (**L-AA 25**) administered at 17 d of incubation (**doi**) on body weight (**BW**) and eye characteristics of broilers at 0, 7, 14, 21, and 28 d of posthatch age (**doa**).

Treatment	BW (g)	AEYW ^1^ (g)	REYW ^1^ (%)
		0 doa	
Non-injected ^2^	47.8	2.13	4.47
Saline ^3^	47.0	2.13	4.55
L-AA 12 ^4^	46.6	2.07	4.45
L-AA 25 ^5^	47.9	1.97	4.10
SEM ^6^	1.08	0.074	0.169
*p*-value	0.546	0.102	0.056
		7 doa	
Non-injected	117	2.23	1.94
Saline	103	2.13	2.11
L-AA 12	113	2.30	2.06
L-AA 25	108	2.13	2.01
SEM	6.6	0.100	0.129
*p*-value	0.201	0.280	0.618
		14 doa	
Non-injected	353	3.50	1.01
Saline	362	3.53	0.99
L-AA 12	395	3.50	0.91
L-AA 25	373	3.40	0.94
SEM	30.2	0.219	0.078
*p*-value	0.543	0.936	0.577
		21doa	
Non-injected	667	4.60	0.71
Saline	742	4.50	0.64
L-AA 12	766	4.73	0.63
L-AA 25	719	4.93	0.71
SEM	63.4	0.280	0.063
*p*-value	0.447	0.453	0.419
		28 doa	
Non-injected	1296	5.87	0.46
Saline	1149	5.77	0.52
L-AA 12	1169	5.43	0.46
L-AA 25	1300	6.00	0.50
SEM	91.0	0.324	0.043
*p*-value	0.344	0.357	0.353

^1^ Absolute eye weight (AEYW) and eye weight relative to BW (REYW); ^2^ Eggs that were not injected.; ^3^ Eggs that were injected with 100 μL saline at 17 doi; ^4^ Eggs that were injected with 100 μL saline containing L-AA 12 at 17 doi.; ^5^ Eggs that were injected with 100 μL saline containing L-AA 25 at 17 doi.; N = 2 birds in each of 12 replicate groups in each treatment combination were used for means calculations.; ^6^ Standard error of the mean.

**Table 6 animals-13-00399-t006:** Effects of treatment [non-injected; saline-injected (saline); saline containing 12 mg of L-ascorbic acid (L-AA 12), or 25 mg of L-ascorbic acid (L-AA 25) administered at 17 d of incubation (doi) on the body weight (BW), and breast meat weight of broilers at 28 d of posthatch age (doa).

Treatment	BW	AP. Major ^1^	AP. Minor ^1^	ABR ^1^	RP. Major ^1^	RP. Minor ^1^	RBR ^1^
	---------------------------g------------------------				--------------------(%)-------------------		
Non-injected ^1^	1296	219	47.1	266	16.7	3.6	20.4
Saline ^2^	1149	187	40.7	228	16.0	3.5	19.5
L-AA 12 ^3^	1169	208	44.4	252	16.7	3.6	20.4
L-AA 25 ^4^	1300	220	47.2	268	17.1	3.7	20.8
SEM ^5^	91.0	23.3	4.15	27.2	0.93	0.15	1.03
*p*-value	0.344	0.478	0.367	0.452	0.705	0.732	0.674

^1^ absolute weights of the pectoralis major (AP. major), pectoralis minor (AP. minor), and breast muscle (ABR), and relative (percentages of BW) weights of the pectoralis major (RP. major) and minor (RP. minor), and breast (RBR); ^2^ Eggs that were not injected.; ^3^ Eggs that were injected with 100 μL saline at 17 doi.; ^4^ Eggs that were injected with 100 μL saline containing L-AA 12 at 17 doi.; ^5^ Eggs that were injected with 100 μL saline containing L-AA 25 at 17 doi.; N = 2 birds in each of 12 replicate groups in each treatment combination were used for means calculations.

**Table 7 animals-13-00399-t007:** Effects of treatment [non-injected; saline-injected (saline); saline containing 12 mg of L-ascorbic acid (L-AA 12), or 25 mg of L-ascorbic acid (L-AA 25) administered at 17 d of incubation (doi) on corneal erosion evaluation scores ^1^ of broilers at 14, 21, 28, and 35 d of posthatch age (doa).

Treatment		Score 0 ^2^	Score 1 ^3^
		------------------(%)------------------	
14 doa			
	Non-injected ^4^	99.9	0.1
	Saline ^5^	97.2	2.8
	L-AA 12 ^6^	95.8	4.2
	L-AA 25 ^7^	100	0
	SEM ^8^	4.4	4.4
21 doa			
	Non-injected	91.7 ^a^	8.3 ^b^
	Saline	73.6 ^b^	26.4 ^a^
	L-AA 12	88.9 ^a^	11.1 ^b^
	L-AA 25	94.4 ^a^	5.6 ^b^
	SEM	4.39	4.39
28 doa			
	Non-injected	93.1	6.9
	Saline	83.3	16.7
	L-AA 12	83.3	16.7
	L-AA 25	90.3	9.7
	SEM	4.39	4.39
35 doa			
	Non-injected	100	0
	Saline	95.8	4.2
	L-AA 12	100	0
	L-AA 25	98.6	1.4
	SEM	4.39	4.30
*p*-values			
	*in ovo*	0.209	0.209
	Day	<0.0001	<0.0001
	*In ovo* x Day	0.003	0.003

^a,b^ Treatment means within the same variable column and within type of treatment with no common superscript differ significantly (*p* < 0.05).; ^1^ No category 2 scores (corneal perforation/lesions with keratoconjunctivitis) were observed and recorded. Therefore, only the score 0 and 1 data are presented.; ^2^ Normal eye exhibiting no ocular abnormality.; ^3^ Signs of ocular inflammation.; ^4^ Eggs that were not injected with no solution.; ^5^ Eggs that were injected with 100 μL saline at 17 d of incubation.; ^6^ Eggs that were injected with 100 μL saline containing L-AA 12 at 17 doi.; ^7^ Eggs that were injected with 100 μL saline containing L-AA 25 at 17 doi.; ^8^ Standard error of the mean. N = 4 birds in each of 12 replicate groups in each treatment combination were used for means calculations.

## Data Availability

None of the data were deposited in an official repository.

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
