# Peer review of "Effects of the In Ovo Administration of L-ascorbic Acid on the Performance and Incidence of Corneal Erosion in Ross 708 Broilers Subjected to Elevated Levels of Atmospheric Ammonia†"

_animals, 2023, doi:10.3390/ani13030399_

Round 1

Reviewer 1 Report

The manuscript provides empirical roles of in ovo injection of L-ascorbic acid to ameliorate the toxicity effects of atmospheric ammonia by studying the performance and corneal erosion of broiler chickens. The use of in ovo technique is especially novel and therefore it is worth publishing as it contributes to the advancement of the science in this area. NH3 exposure is inevitable in broiler housing and this research can be practically useful to be applied to the broiler industry.

The manuscript is well-written and well presented. The objective was clearly stated in the introduction which supported by sufficient theoretical background and gap of knowledge. However, the authors need to state the hypothesis they tested in this experiment.

The materials and methodology are clear, presented in chronological order with sufficient details of information are also available. They are all matched with the results they presented which are also wellpresented.

Statistical analysis: Correct!

All values presented in the results are make sense and ranged in the expected values.

Discussion: sufficient, but I suggest to explain why 25 mg L-AA did not improve the performance compared to 12 mg L-AA?

Conclusion: authors need to make the conclusion consistent between summary, abstract, and conclusion section, especially related to the 12/25 mg L-AA treatment. I suggest the author to revise the conclusion section to just retain the 12 mg L-AA treatment that was able to improve the performance and corneal erosion incidence.

Other suggestion: Some references are outdated, for example see Ref. no. 2,3, 6, 7 etc. Authors may consider to use the newer references as there are many progress of research about the role of Vit. C in poultry. 

This study provides novelty in this area. I enjoyed reading the manuscript, it is well written with clear chronological order and well methodological descriptions.

I have few additional suggestions that may be considered by authors: 

L21: NH3 -> NH(please check thoroughly)

L23: In ovo -> italic

L61: in vitro -> not sure if it should be italic (check the journal style requirement)

Consistency of table formatting (i.e., Table 4)

 Results:

Not sure if there is a requirement to include p-value in the text, I am ok with both, but please check the guideline. Be consistent since you put p-value in other subsection (3.2).

Conclusion:

- L440-441: 25 mg of in ovo L-AA treatment did not show an improvement on performance (see Table 4, it is not significantly different with saline control), so this statement is not completely true. Please adjust accordingly if you think it is accurate.  

- L443-444: it is too speculative for conclusion since none of the immunity, antioxidant, and inflammatory parameters were analyzed in this study. You may say that it is an important point to analyze those parameters to confirm the L-AA effect. 

Author Response

Reviewer 1:

The manuscript provides empirical roles of in ovo injection of L-ascorbic acid to ameliorate the toxicity effects of atmospheric ammonia by studying the performance and corneal erosion of broiler chickens. The use of in ovo technique is especially novel and therefore it is worth publishing as it contributes to the advancement of the science in this area. NH3 exposure is inevitable in broiler housing and this research can be practically useful to be applied to the broiler industry.

The manuscript is well-written and well presented. The objective was clearly stated in the introduction which supported by sufficient theoretical background and gap of knowledge. However, the authors need to state the hypothesis they tested in this experiment.

Answer:

Thank you for the comments and suggestion. The following hypothesis statement was inserted in the introduction section on lines 104-106; “it is hypothesized that some level of the in ovo injection of L-AA would lower the negative effects of chronic aerial ammonia exposure on the live performance of broilers”.  

The materials and methodology are clear, presented in chronological order with sufficient details of information are also available. They are all matched with the results they presented which are also well presented.

Statistical analysis: Correct!

All values presented in the results are make sense and ranged in the expected values.

Discussion: sufficient, but I suggest to explain why 25 mg L-AA did not improve the performance compared to 12 mg L-AA?

Answer:

Thank you for the suggestion, the relevant correction was added to discussion section on line 447-454.

“Posthatch performance responded more favorably to the in ovo injection of 12 mg of L-AA rather than to 25 mg of L-AA. Similar results were observed when L-AA above 12 mg was administrated by in ovo injection. It was observed that the in ovo injection of 36 mg of L-AA negatively affected the BW and FCR of broilers in comparison to 12 mg L-AA and saline control groups [37]. Zhang et al. [37] suggested that the higher level of L-AA could lead to an inferior feed efficiency or to a subsequently poorer utilization of feed. Thus, this could be a partial reason for the unexpected response to the in ovo injection of 25 mg of L-AA. ”

Conclusion: authors need to make the conclusion consistent between summary, abstract, and conclusion section, especially related to the 12/25 mg L-AA treatment. I suggest the author to revise the conclusion section to just retain the 12 mg L-AA treatment that was able to improve the performance and corneal erosion incidence.

Answer:

Thank you for the suggestion. The relevant correction was added to the discussion section.

Other suggestion: Some references are outdated, for example see Ref. no. 2,3, 6, 7 etc. Authors may consider to use the newer references as there are many progress of research about the role of Vit. C in poultry. 

Answer:

The more updated references replaced the old references and the relevant corrections were applied in the text.

This study provides novelty in this area. I enjoyed reading the manuscript, it is well written with clear chronological order and well methodological descriptions.

Answer:

Thank you for the comments.

I have few additional suggestions that may be considered by authors: 

L21: NH3 -> NH(please check thoroughly)

Answer:

The relevant corrections were applied in the text.

L23: In ovo -> italic

Answer:

We would like to make “in ovo italicized; however, the Animals journal did not allow us to do that in the past.

L61: in vitro -> not sure if it should be italic (check the journal style requirement)

Answer:

We would like to make “in vitro as well as in ovo italicized; however, the Animals journal did not allow us to do that in the past.

Consistency of table formatting (i.e., Table 4)

Answer:

Thank you for the comment. We made the treatment description equal in all relevant tables, and made some other adjustments as well. Please let us know if there is any specific inconsistency that we need to address properly. 

 Results:

Not sure if there is a requirement to include p-value in the text, I am ok with both, but please check the guideline. Be consistent since you put p-value in other subsection (3.2).

Answer:

Thank you for the comment. There are no specific journal’s guidelines for the inclusion of P-values in the text. The P-values in the 3.2 subsection were removed.

Conclusion:

- L440-441: 25 mg of in ovo L-AA treatment did not show an improvement on performance (see Table 4, it is not significantly different with saline control), so this statement is not completely true. Please adjust accordingly if you think it is accurate.  

 Answer:

We agree that the 25 mg dosage did not result in a better performance in comparison to the control group. Thus, we removed it from that statement.

- L443-444: it is too speculative for conclusion since none of the immunity, antioxidant, and inflammatory parameters were analyzed in this study. You may say that it is an important point to analyze those parameters to confirm the L-AA effect. 

 Answer:

In our previous research, we found that the in ovo injection of L-AA resulted in an improvement in the inflammatory reaction and antioxidant activity of grown broilers when the same treatment was used. These statements are currently presented on Lines 390-392 (immunity), and lines 408-412 and 426-429 (antioxidant activity). In addition, we also suggested that these measurements needed to be measured in the future research in the conclusion section. 

Reviewer 2 Report

GENERAL COMMENT:

I consider this work is within the scope of “Animals”. It contains information useful in a field in which available information is scarce and of interest to improve knowledge on broiler rearing. Overall, it is well written and organised. I indicate below only minor points to be improved in the manuscript.

ACROSS THE ENTIRE MANUSCRIPT:

Type "3" as a subscript in “NH3”, thus resulting in “NH3

ABSTRACT:

It is OK.

KEYWORDS:

Line 52: I suggest adding “broiler” or “chicken” as key word.

INTRODUCTION:

This section is OK.

Line 106: Add “Ross 708” thus resulting in “erosion in Ross 708 broilers exposed to”.

MATERIALS AND METHODS:

Line 118: Add average relative humidity during the different incubation stages.

Lines 116-119: Describe egg turning frequency and duration in the incubator.

Line 117: Incubation temperature was maintained constant at 37.5 ºC until hatch?

Line 160: Insert space at “to7”, thus resulting in “to 7”.

RESULTS SECTION:

This section is OK.

DISCUSSION SECTION:

This section is OK.

REFERENCES SECTION:

In general terms, this section is well organised and adjusted to the style of the journal for references. However, I recommend reviewing it for typos.

TABLES:

Table 1: Type “Kcal” with lowercase letter: “kcal”. According to SI of units, k from kilo is always written as lowercase letter.

Table 3: Indicate in the footnote what it means “SEM”.

Table 4: Indicate in the footnote what it means “SEM”.

Table 5: Indicate in the footnote what it means “SEM”.

Table 6: Indicate in the footnote what it means “SEM”.

Author Response

Reviewer 2

I consider this work is within the scope of “Animals”. It contains information useful in a field in which available information is scarce and of interest to improve knowledge on broiler rearing. Overall, it is well written and organised. I indicate below only minor points to be improved in the manuscript.

ACROSS THE ENTIRE MANUSCRIPT:

Type "3" as a subscript in “NH3”, thus resulting in “NH3

 Answer:

The relevant corrections were applied to the text.

ABSTRACT:

It is OK.

KEYWORDS:

Line 52: I suggest adding “broiler” or “chicken” as key word.

 Answer:

The relevant correction was applied to the text.

INTRODUCTION:

This section is OK.

Line 106: Add “Ross 708” thus resulting in “erosion in Ross 708 broilers exposed to”.

 Answer:

The relevant correction was applied to the text.

MATERIALS AND METHODS:

Line 118: Add average relative humidity during the different incubation stages.

 Answer:

Thank you for the comments. We provided dry and wet bulb temperatures, which is a common practice in incubational papers. It is common practice to calculate relative humidity from the dry and wet bulb temperatures. The approximate relative humidity for this study was 55%, which was also employed in the reference related to this section.

Lines 116-119: Describe egg turning frequency and duration in the incubator.

 Answer:

The relevant correction was applied to the text on lines 123 and 124.

Line 117: Incubation temperature was maintained constant at 37.5 ºC until hatch?

 Answer:

There was a slight difference in setter and hatcher temperature. The relevant corrections were applied to the text on lines 120-122.

“set in the setter at 37.5 ºC dry bulb and 29.0 ºC wet bulb temperatures and in the hatcher at 36.9 ºC dry bulb and 29.0 ºC wet bulb temperatures.”

Line 160: Insert space at “to7”, thus resulting in “to 7”.

 Answer:

The relevant correction was applied to the text.

RESULTS SECTION:

This section is OK.

DISCUSSION SECTION:

This section is OK.

REFERENCES SECTION:

In general terms, this section is well organised and adjusted to the style of the journal for references. However, I recommend reviewing it for typos.

TABLES:

Table 1: Type “Kcal” with lowercase letter: “kcal”. According to SI of units, k from kilo is always written as lowercase letter.

 Answer:

The relevant correction was applied to the text.

Table 3: Indicate in the footnote what it means “SEM”.

 Answer:

The relevant correction was applied to Table 3.

Table 4: Indicate in the footnote what it means “SEM”.

 Answer:

The relevant correction was applied to Table 4.

Table 5: Indicate in the footnote what it means “SEM”.

 Answer:

The relevant correction was applied to Table 5.

Table 6: Indicate in the footnote what it means “SEM”.

 Answer:

The relevant correction was applied to Table 6.

Reviewer 3 Report

Simple summary, Abstract:  You can specify control groups as negative and positive separately.

Lines 61-62:   You can remove this part as it is not directly related to the subject.   “Siegel [8] has documented that L-AA has direct in vitro virucidal and bactericidal activity against a number of pathogens such as Newcastle disease”.

Line 112: please delete “fertile”. simply enter "broiler hatching eggs". It's possible that the initial fertilization of the eggs is unknown.

Line 113: “under commercial conditions”?, Please specify the storage temperature and humidity.

Line 119-120: Why was the fertility control done on the 12th day of incubation?

Lines 121-125: Why were previous weight losses determined for an application performed on the 17th day of incubation? What kind of advantage will the weight loss in this process provide?

Lines 127-133: How did you determine the application doses?

Lines 133-134: For the preparation of the solution, it is not enough to just cite; it would be more appropriate to give brief information. Please provide specifics for all procedures. 

Table 7 shows the averages for the main effects, followed by the averages for the interaction effects. Looking at the results of the statistical analysis, it is seen that the interaction effect is significant. In this case, what needs to be done is to reveal the differences for each subgroup mean with an appropriate post-hoc test. However, in your study, only post-hoc results were given for the "21 doa * in-ovo" interaction. In multiway-ANOVA or MANOVA analyzes like this, when the interaction effects are significant, a new dummy variable can be created and then you can combine your variables and transform them into one-way. For example, for variables X(1,2) and Y(a,b), the dummy variable could be XY (1a, 1b, 2a, 2b). Re-analyze for the interaction effects in Table 7 and redo your discussion accordingly.

Lines 438-448: You cannot say this sentence. Because there is no statistical difference between your application groups and the control group. There are only numerical differences.  “The results of the current study revealed that the in ovo injection of 12 or 25 mg of L-AA 440 resulted in an improvement in broiler performance from 0 to 28 doa” Please review the conclusion and abstract part. In your study, there is no statistical difference between the control group and the application groups in terms of performance and carcass in the 0-28 or 0-35-day period. 

Author Response

Reviewer 3

Simple summary, Abstract:  You can specify control groups as negative and positive separately.

 Answer:

Thank you for the suggestion. As levels of inclusion, positive and negative controls are referred to as being absent and present, respectively. However, in this study, we did not have actual positive control. Saline cannot be considered as a positive control, because the L-AA treatments also contained similar amounts of saline.

Lines 61-62:   You can remove this part as it is not directly related to the subject.   “Siegel [8] has documented that L-AA has direct in vitro virucidal and bactericidal activity against a number of pathogens such as Newcastle disease”.

 Answer:

The aforementioned sentence was removed.

Line 112: please delete “fertile”. simply enter "broiler hatching eggs". It's possible that the initial fertilization of the eggs is unknown.

 Answer:

The word “fertile” was removed from the sentence on line 113. 

Line 113: “under commercial conditions”?, Please specify the storage temperature and humidity.

 Answer:

The relevant correction was inserted to lines 115-116. The correct sentence is as follows: “12.8 ºC and 10.4 ºC dry and wet bulb temperatures, respectively”

Line 119-120: Why was the fertility control done on the 12th day of incubation?

 Answer:

The eggs were candled to remove infertile and early dead mortalities.

Lines 121-125: Why were previous weight losses determined for an application performed on the 17th day of incubation? What kind of advantage will the weight loss in this process provide?

 Answer:

Percentage egg weight loss indicates the amount of water that is lost from the egg during the incubation period and reflects the incubational conditions (temperature and humidity). When there is no significant difference between treatments for this variable, it means that they experienced similar incubational conditions, and the possible differences observed at hatch and posthatch are not confounded by the incubational conditions. 

Lines 127-133: How did you determine the application doses?

 Answer:

This research is the continuation of several previous studies in our laboratory, in which we found that these levels of L-AA may have a greater potential for use in the current study.

The pervious researches are listed below:

  1. Zhang, H.; Elliott, K.E.C.; Durojaye, O.A.; Fatemi, S.A.; Peebles, E.D. Effects of in ovo-administration of L-ascorbic acid on ‎broiler hatchability and its influence on the effects of pre-placement holding time on broiler quality characteristics. Poult. ‎ 2018, 97, 1941–1947.‎
  2. Mousstaaid, A.; Fatemi, S.A.; Elliott, K.E.C.; Alqhtani, A.H.; Peebles, E.D. Effects of the in ovo injection of L-ascorbic acid on ‎broiler hatching performance. Animals Basel 2022, 12, 1020. https://doi.org/10.3390/ani12081020.‎
  3. ‎Mousstaaid, A.; Fatemi, S.A.; Elliott, K.E.C.; Levy, A.W.; Miller, W.W.; Gerard P.D.; Alqhtani, A.H.; Peebles, E.D. Effects of ‎the in ovo and dietary supplementation of L-ascorbic acid on the growth performance, inflammatory response, and eye ‎L-ascorbic acid concentrations in Ross 708 broiler chickens. Animals Basel 2022. 12, 2573. https://doi.org/10.3390/ani12192573.‎
  4. Zhang, H.; Elliott, K.E.C.; Durojaye, O.A.; Fatemi, S.A.; Schilling, M.W.; Peebles, E.D. Effects of in ovo injection of L-ascorbic acid on growth performance, carcass composition, plasma antioxidant capacity, and meat quality in broiler chickens. Poult. Sci. 2019, 98, 3617–3625.

Lines 133-134: For the preparation of the solution, it is not enough to just cite; it would be more appropriate to give brief information. Please provide specifics for all procedures. 

 Answer:

The additional information was added to the Materials and Methods section on line 132.

Table 7 shows the averages for the main effects, followed by the averages for the interaction effects. Looking at the results of the statistical analysis, it is seen that the interaction effect is significant. In this case, what needs to be done is to reveal the differences for each subgroup mean with an appropriate post-hoc test. However, in your study, only post-hoc results were given for the "21 doa * in-ovo" interaction. In multiway-ANOVA or MANOVA analyzes like this, when the interaction effects are significant, a new dummy variable can be created and then you can combine your variables and transform them into one-way. For example, for variables X(1,2) and Y(a,b), the dummy variable could be XY (1a, 1b, 2a, 2b). Re-analyze for the interaction effects in Table 7 and redo your discussion accordingly.

 Answer:

Thank you for the comment. Because significant interaction effects supersede main effects, the main effect means in Table 7 were removed, and only the doa x in ovo treatment interaction subgroup means are provided. Accordingly, any mention or discussion of main effect differences was deleted from Results and Discussion section. Only, in ovo treatment effect within 21 doa was addressed. The two-way repeated measure ANOVA was appropriate analysis that allowed for this interaction to be revealed and to reported and discussed.

Lines 438-448: You cannot say this sentence. Because there is no statistical difference between your application groups and the control group. There are only numerical differences.  “The results of the current study revealed that the in ovo injection of 12 or 25 mg of L-AA 440 resulted in an improvement in broiler performance from 0 to 28 doa” Please review the conclusion and abstract part. In your study, there is no statistical difference between the control group and the application groups in terms of performance and carcass in the 0-28 or 0-35-day period. 

 Answer:

We revised the sentence and included only the 12 mg of L-AA treatment when an improvement was observed for BWG, ADG and FCR from 0 to 28 doa in comparison to the saline control group.

Round 2

Reviewer 3 Report

Dear Authors,

For future studies,
If the interaction effect is significant in the tables, interaction lettering should be done and discussed.
It may not be appropriate for you to cite too much yourself, even if it relates to the current study.

Best regards,